# Characterization of terminal flowering cowpea (*Vigna unguiculata* (L.) Walp.) mutants obtained by induced mutagenesis digs out the loss-of-function of phosphatidylethanolamine-binding protein

Vijayakumar Eswaramoorthy[1], Thangaraj Kandasamy[2]*, Kalaimagal Thiyagarajan[1]*, Vanniarajan Chockalingam[2], Souframanien Jegadeesan[3], Senthil Natesan[4], Karthikeyan Adhimoolam[5], Jeyakumar Prabhakaran[6], Ramji Singh[7], Raveendran Muthurajan[4]

1 Department of Plant Breeding and Genetics, Agricultural College and Research Institute, Tamil Nadu Agricultural University, Coimbatore, Tamil Nadu, India, 2 Department of Plant Breeding and Genetics, Agricultural College and Research Institute, Tamil Nadu Agricultural University, Madurai, Tamil Nadu, India, 3 Nuclear Agriculture & Biotechnology Division, Bhabha Atomic Research Centre, Trombay, Mumbai, India, 4 Centre for Plant Molecular Biology and Biotechnology, Tamil Nadu Agricultural University, Coimbatore, India, 5 Subtropical Horticulture Research Institute, Jeju National University, Jeju, South Korea, 6 Department of Crop Physiology, Agricultural College and Research Institute, Tamil Nadu Agricultural University, Coimbatore, Tamil Nadu, India, 7 Department of Plant Pathology, College of Agriculture, Sardar Vallabhbhai Patel University of Agricultural Sciences and Technology, Meerut, Uttar Pradesh, India

* ka.thangaraj@gmail.com (TK); t.kalaimagal@gmail.com (KT)

## Abstract

Cowpea (*Vigna unguiculata* (L.) Walp) is one of the major food legume crops grown extensively in arid and semi-arid regions of the world. The determinate habit of cowpea has many advantages over the indeterminate and is well adapted to modern farming systems. Mutation breeding is an active research area to develop the determinate habit of cowpea. The present study aimed to develop new determinate habit mutants with terminal flowering (TFL) in locally well-adapted genetic backgrounds. Consequently, the seeds of popular cowpea cv P152 were irradiated with doses of gamma rays (200, 250, and, 300 Gy), and the $M_1$ populations were grown. The $M_2$ populations were produced from the $M_1$ progenies and selected determinate mutants (TFLCM-1 and TFLCM-2) from the $M_2$ generation (200 Gy) were forwarded up to the $M_5$ generation to characterize the mutants and simultaneously they were crossed with P152 to develop a MutMap population. In the M5 generation, determinate mutants (80–81 days) were characterized by evaluating the TFL growth habit, longer peduncles (30.75–31.45 cm), erect pods (160˚- 200˚), number of pods per cluster (4–5 nos.), and early maturity. Further, sequencing analysis of the *VuTFL1* gene in the determinate mutants and MutMap population revealed a single nucleotide transversion (A-T at 1196 bp) in the fourth exon and asparagine (N) to tyrosine (Y) amino acid change at the $143^{rd}$ position of phosphatidylethanolamine-binding protein (PEBP). Notably, the loss of function PEPB with a higher confidence level modification of anti-parallel beta-sheets and destabilization of the protein secondary structure was observed in the mutant lines.

**Data Availability Statement:** All relevant data are within the paper and its Supporting information files.

**Funding:** Research work was supported by the "Board of Research in Nuclear Sciences (BRNS)" by the Bhabha Atomic Research Centre (BARC), Trombay, India. Grant number: No.35/14/18/2018-BRNS/10396 dt. 25.05.2018." The funders had no role in study design, data collection and analysis, decision to publish, or preparation of the manuscript.

**Competing interests:** The authors have declared that no competing interests exist.

Quantitative real-time PCR (qRT-PCR) analysis showed that the *VuTFL1* gene was downregulated at the flowering stage in TFL mutants. Collectively, the insights garnered from this study affirm the effectiveness of induced mutation in modifying the plant's ideotype. The TFL mutants developed during this investigation have the potential to serve as a valuable resource for fostering determinate traits in future cowpea breeding programs and pave the way for mechanical harvesting.

## Introduction

Grain legumes have been an affordable and sustainable source of human dietary protein for centuries. Owing to its potential, the Food and Agriculture Organization of the United Nations [1] called them "Nutritious seeds for a sustainable future". Cowpea (*Vigna unguiculata* (L.) Walp) is a nutritionally secure and climate-resilient crop as it supplies a plethora of nutrients at a low cost and has the ability to grow in unforgiving and harsh climatic conditions by fixing atmospheric nitrogen in the soil [2]. It is mainly cultivated by marginal farmers as rice fallow pulses. Despite its versatility and relevance to agriculture, cowpea productivity is very low, and the least importance is given to its production technology. A steep reduction in productivity is mainly due to its indeterminate twining growth habits and asynchronous maturity [3]. The prostrate growth habit of cowpea deviates most of its energy in the vegetative phase, leaving less photosynthates for the sink [4]. Furthermore, it requires more spacing than other pulses and makes intercultural operations a challenging task. At the time of harvest, half of the pods remain immature due to asynchronous maturity. This necessitates two or more pickings for complete harvest and requires a large number of labours. Cowpea possesses genetic variation in terms of growth habit types and it has been classified as indeterminate or determinate habit depending on whether the terminal meristems are vegetative or reproductive. The determinate habit of cowpea has many advantages over the indeterminate including early flowering, synchronised maturity, photo insensitivity, ease of mechanical harvesting and intercultural operations, and, suitable for modern farming systems. Therefore, it is necessary to develop a cowpea line with high yield, determinate growth pattern/ terminal flowering (TFL), erect growth habit, and branches with pods above the canopy of a plant is necessary. Moreover, altering the ideotype of cowpea from indeterminate to determinate types would enhance its adaptability and stability towards climate change [5,6] and to address labour shortage problems by its suitability for mechanical harvest [7].

The life cycle of cowpea is bifurcated into vegetative and reproductive growth phases. The transition from one phase to the other is controlled by a series of genetic pathways and physiological signals [8,9]. Shoot apical meristem (SAM) plays a key role in the transition from the vegetative to reproductive phase [10]. In the determinate types, SAM will be transmuted to terminal floral meristem and it arrests the vegetative growth, contrarily SAM continues to grow and extends the vegetative growth in indeterminate types [11]. TFL is controlled by the highly conserved *TFL1* (TERMINAL FLOWER 1)/*CEN* (CENTRORADIALIS) gene [12,13] that consists of four exons and three introns and is located in chromosome 1 [5,14]. *TFL1* plays a diverse role in signaling pathways regulating growth and differentiation [15]. It also helps in maintaining floral architecture, flowering time, and self-pruning habits [16,17]. *TFL1* and FLOWERING LOCUS T (*FT*) are the two genes from the PEBP family which determine the flowering of the plants [18]. *FT* and *TFL1* genes act oppositely while regulating the flowering. *FT* converts SAM to a flower thus promoting flowering [19] whereas *TFL1* prolongs the SAM and delays the up-regulation of the flowering gene *FT* resulting in indeterminate growth habit

[20]. A mutation in *TFL1* with a functional loss is expected to result in the conversion of apical meristem into terminal flower [21,22]. Ahn, Miller [23] reported that a functional mutation in the fourth exon provides greater plasticity for the conversion of indeterminate to determinate plant types. Among the different approaches deployed for altering the ideotype of the plant, gamma-ray induced mutations were found to be effective and its feasibility in inducing the TFL in cowpea has been previously manifested by Pandey and Dhanasekar [24] and Dhanasekar and Reddy [5].

Mutation breeding is an active research area to develop the determinate habit of cowpea. MutMap combines mutation breeding with hybridization to construct the mapping population at the mutant loci. This method targets the single nucleotide polymorphism (SNP) responsible for phenotypic modification which is procreated through direct mutagenesis and relies upon the cross between the mutant type and its wild type [25]. MutMap technique was first deployed by Abe, Kosugi [26] in rice for mapping traits that are regulated by monogenic recessive genes. Previous reports in soybean, *Arabidopsis*, and rose suggested that *TFL1* is a homozygous recessive gene [27,28]. In light of the above facts, the present study aimed at identifying the new determinate habit mutants with TFL by mutation breeding approach. For this purpose, we developed the $M_1$-$M_5$ mutants and MutMap population and identified the new determinate habit mutants with TFL. In addition, we have also characterized the mutants by sequence and expression analyses. The newly identified cowpea mutants from this research are useful genetic stocks for genetic improvement and breeding.

## Materials and methods

### Plant genetic materials and experimental site

The cowpea cultivar P152 (Selection from P42) is an indeterminate growth type (90 to 95 days duration) with puff coloured seed. It is predominantly cultivated in North and South India, especially in Tamil Nadu. The P152 seeds were obtained from the Department of Plant genetic resources, Centre for Plant Breeding and Genetics, Tamil Nadu Agricultural University, Coimbatore, Tamil Nadu, India. All the field experiments were conducted at the experimental farm (9˚ 54″ N; 78˚ 54″ E) from 2018 to 2021 at Agricultural College and Research Institute, Tamil Nadu Agricultural University, Madurai, India.

### Mutagenic dosage optimization

One thousand well-filled and uniform seeds of P152 cowpea were selected and equilibrated to 12% moisture content. For treatment, two hundred seeds per dose were taken in five butter paper covers and irradiated separately with 150, 200, 250, 300, and, 350 Gy of gamma rays using a gamma chamber (Model GC 5000, BRIT, India) installed at the Bhabha Atomic Research Centre, Trombay, Mumbai, India. Cobalt 60 ($^{60}$Co) was used for sourcing the gamma rays. The seeds were irradiated at the dose rate of 24.5 Gy/min and the treatment durations were 6.3 minutes for 150 Gy, 8.16 minutes for 200 Gy, 10.20 minutes for 250 Gy, 12.25 minutes for 300 Gy and14.3 minutes for 350 Gy to get respective doses. The treated seeds were sown along with control (untreated P152 seeds) in roll-towel, pro tray, and field to find out the optimal dose for the cowpea variety under study. A total of fifty seeds from each mutagenic treatment were sown in two replications. Germination percentage was recorded on the seventh day after sowing in all mutagenic treatments including control. Probit analysis [29] was carried out with estimates of germination percentage on the seventh day after sowing for determination of lethal dose ($LD_{50}$) value. Based on the probit analysis, the doses viz., 200, 250, and, 300 Gy were regarded as appropriate doses for induced mutagenesis.

## Induced mutagenesis, selection, and assessment of mutants

Two thousand hand-picked seeds of P152 per each optimal dose of gamma rays (200, 250, and, 300 Gy) were treated, and the seeds were allowed to germinate in the fields of Agricultural College and Research Institute, Madurai, India during Rabi 2018 (October 2018 to January 2019) along with the control (Non-mutagenized P152 wild-type plants). A single seed per hill was sown with a spacing of $30 \times 10$ cm in three-meter rows. The recommended package of practices was followed to raise the healthy crops as outlined in the crop production guide (CPG) for Tamil Nadu (TNAU, 2018). All the survived $M_1$ plants were selfed and the seeds were used to raise the $M_2$ generation. The terminal mutants were identified and forwarded to $M_3$ generation along with the wild type (P152). The observations on 11 morphological traits on $M_3$ generation viz., plant height (cm), days to flowering, number of primary branches, number of clusters per plant, number of pods per plant, peduncle length (cm), pod length (cm), number of seeds per pod, hundred seed weight (g), days to maturity, and single plant yield (g) were recorded on randomly 500 plants per dose. The selected mutants were forwarded to $M_4$ and $M_5$ generations to check the stability of the mutants during Kharif 2020 (June 2020 to August 2020) and Rabi 2020 (October 2020 to January 2021).

## Development of MutMap population

The TFL mutants identified in the $M_2$ generation of gamma rays (200 Gy) were used as females and were crossed with the wild type as male parent (P 152) to produce the mutant filial generation 1 ($MF_1$). The $MF_1$ generation (90 plants) was selfed to produce modified MutMap population/mutant filial generation 2 ($MF_2$). The observations on the segregating pattern of the MutMap population were recorded. A brief layout for the development of TFL mutants and MutMap population is provided in S1 Fig. The observations on 14 morphological traits viz., plant height (cm), days to flowering, mean internodal length (cm), number of leaves, number of primary branches, number of clusters per plant, number of pods per plant, peduncle length (cm), pod length (cm), number of pods per cluster, number of seeds per pod, hundred seed weight (g), days to maturity, and single plant yield (g) were recorded on ten randomly selected TFL plants.

## Statistical analysis

The morphological data obtained from mutants of $M_2$ generation along with wild type were subjected to variability analysis viz., genotypic and phenotypic coefficient of variation, heritability, and genetic advance were calculated following standard methods [30–33]. The data analysis was carried out using the TNAUSTAT software [34]. In MutMap/$MF_2$ generation, the segregation patterns of mutants and wild type were analysed by chi-square test [35].

## Amplification and sequencing of *VuTFL1* in selected mutants

Young leaves from TFL mutants and wild type were used for genomic DNA isolation using the modified cetyltrimethylammonium bromide (CTAB) method as suggested by Rogers and Bendich [36]. For TFLCM-MF2 and IDCM-2, the DNA's of randomly selected 32 determinate and indeterminate plants, respectively, were pooled in equimolar concentration. Quality and quantity of genomic DNA were ascertained by resolving on 0.8% agarose gel and Nano-Drop1000 spectrophotometer (Thermo Scientific, US), respectively. DNA samples were diluted to 50 ng/µL concentration with ddH2O and stored at−20 ˚C. The gene-specific primers were designed based on the 5′-untranslated regions (UTR) and 3′-UTR of the cowpea terminal flowering gene (*VuTFL1*) sequences retrieved from the NCBI (Accession number: KJ569520).

**Table 1. List of primers used for PCR and quantitative real-time PCR analyses.**

| S. No | Gene Name | Primer set | Amplicon size (bp) |
|---|---|---|---|
| 1. | *VuTFL1* | F: ATGGCAAGAATGCCTTTAGAAC<br>R: CTAGCGTCTTCTTGCAGCTG | 1291 |
| 2. | RT-*Vu* TFL1 | F: GATGTTCCAGGCCCTAGTGA<br>R: TGTTTGATGCGATGAAAGGA<br>RC: TCCTTTCATCGCATCAAACA | 234 |
| 3. | RT-*Vu* TFL2 | F: GGGAAAGAGTTGGTGAGCTA<br>R: CTCGTTCTGTGCTGCGAAAT<br>RC: ATTTCGCAGCACAGAACGAG | 151 |
| 4. | RT-*Vu* ACT2 | F: TCAGGTGTCCAGAGGTGTTGTA<br>R: ATGGTTGTGCCTCCTGAAAGTA | 151 |

The primer pair were designed using NCBI primer- BLAST Primer3 software (https://primer3.ut.ee/) [37] (Table 1). The full-length sequences of the *VuTFL1* gene were amplified using DNA from TFL mutants, indeterminate mutants, and wild-type plants. Briefly, PCR was conducted in a total volume of 10μl, including 2.0 μl of genomic DNA, 0.5 μl of the forward primer, and 0.5 μl of reverse primer (Eurofins Genomics India, Bangalore, India), 1X master mix (Amplicon, Denmark) and 1 μl of ddH$_2$O. Amplification was carried out in a thermal cycler (Eppendorf, Hamburg, Germany) programmed to run the following temperature profile: initial denaturation at 94 ˚C for 4 min followed by 35 cycles of 94 ˚C for 30 sec, 56 ˚C for 1 min, 72 ˚C for 2 min, and final extension at 72 ˚C for 10 min. The amplified products were resolved on 1.0% agarose gels using a horizontal gel electrophoresis system (Bio-Rad, California, USA) at 100 V and documented using a gel documentation system (Bio-Rad, California, USA). PCR products were eluted and purified using QIAquick Gel Extraction Kit (Addgene). The purified product was cloned in the binary vector pGEM®T and transformed into the *E. coli* DH5α cells. After the standard blue-white screening, positive colonies were used for plasmid isolation by GenElute TM Plasmid Miniprep Kit (Sigma- Aldrich, USA). Then, the isolated plasmids were used for sequencing (Yaazh Xenomics, Coimbatore, India). All the obtained sequences from this study were submitted to the national centre for biotechnology information (NCBI)/GenBank data libraries with accession numbers OM455488 (P152), OM455489 (IDCM-1), OM455490 (IDCM-2), OM455491 (TFLCM-1-5), OM455492 (TFLCM-2-5) and OM455493 (TFLCM-MF2).

### *In silico* annotation of the sequence data

The obtained genic DNA sequences were aligned using CLUSTAL OMEGA (https://www.ebi.ac.uk/Tools/msa/clustalo/) [38] to identify the changes in nucleotide sequence. The control (P 152) nucleotide sequence was used to predict the gene and demarcate the introns and exons. The *in silico* tool Softberry's FGENESH (http://www.softberry.com/berry.phtml?topic=fgenesh&group=programs&subgroup=gfind) [39] was used to predict the amino acid sequence of all the six lines and were then aligned with CLUSTAL OMEGA to identify the amino acid change. The effect of change in the amino acid sequence was predicted using various *in silico* computational tools viz., PhD-SNP (https://snps.biofold.org/phd-snp/phd-snp.html) [40], and PerdictSNP (https://loschmidt.chemi.muni.cz/predictsnp1/) [41]. The protein annotation viz., protein family, biological process, and molecular function of the gene was predicted using the tool KEGG orthology (https://www.genome.jp/kegg/annotation/) [42]. For protein structure prediction, wild and mutant sequences were subjected to homology

modelling and a template was selected using PDB BLAST (https://blast.ncbi.nlm.nih.gov/Blast.cgi?PAGE=Proteins).

## Protein model building using Accelrys discovery studio 2.5

The amino acid sequence of *VuTFL1* was retrieved from the UniProtKB database in FASTA format and submitted to PDB BLAST to identify suitable templates. The template with maximum identity (E-value) was selected for generating the 3D structure of the target using Modeller in Accelrys discovery studio 2.5 [43]. The per cent identity of alignment was measured as a ratio of the identical residues in the alignment and the total number of residues in the target. The per cent similarity was measured as a ratio of the total number of identical and similar residues to the total number of residues of the target. The modelled 3D structures were visualized using Rasmol (http://www.openrasmol.org/software/rasmol/) [44] and SAVES 6.0 structure validation server (https://saves.mbi.ucla.edu/) was used to validate the predicted protein model. Backbone conformation of each amino acid residue and stereochemical quality of protein models were verified using PROCHECK [45] and SWISS-MODEL Expasy (https://swissmodel.expasy.org/) [46] through Ramachandran plot.

## Quantitative real-time PCR (qRT-PCR) analysis

The *VuTFL1* gene expression was analysed in two determinate mutants (TFLCM-1-5 and TFLCM-2-5) and indeterminate P152 using qRT-PCR analysis. Tissues from the apical bud were collected at four growth stages viz., V2 (Primary leaves), V4 (Third trifoliate leaf), R5 (Pre flowering), and R7 (Pod formation stage) and immediately frozen into liquid nitrogen and then stored at −80˚C [28]. Total RNA was isolated using an RNeasyplant mini kit (Qiagen, Hilden, Germany) and treated with RNase-free DNAseI (Promega, Madison, WI, USA) following manufacture guidelines. The purity and concentration of RNAs were determined using the bio spectrometer (Eppendorf, Hamburg, Germany). According to the manufacturer's instructions, the first-strand cDNA was synthesized using the Thermo Scientific Revert Aid First Strand cDNA synthesis kit (Thermo Scientific, USA). Intron-spanning gene-specific primers for the *VuTFL1* gene were designed using the GenScript Real-time PCR Primer Design tool with default parameters (https://www.genscript.com/ssl-bin/app/primer) (Table 1). Each reaction mixture with a final volume of 10 μl consists of 2 μl of 5X diluted cDNA, 1 μl of each primer at 10 pmol/μl concentration, and 5μl of TB Green® premix Ex Taq™ II as an amplification dye (Tli RNaseH Plus, Cat # RR820A; Takara clone tech) and 1 μl of double-distilled sterile water. All reactions were performed in 96-well plates using a Light Cycler 96 (Roche Diagnostics GmbH, Manheim, Germany) with three technical replicates. The thermal conditions are as follows: initial denaturation for 2 min at 94 ºC and 40 cycles of 10 s at 94 ºC, 30 s at 56 ºC. After amplification, the fluorescence signals were measured at every 0.5 ºC and a melting curve was produced with a gradual rise from 60 ºC to 95 ºC. Actin gene (Internal control) from cowpea was used to normalize, and transcripts change was calculated using the $2^{-\Delta\Delta Ct}$ method [47].

## Results

### Mutagenic dosage optimization

Cowpea cv P152 was irradiated with five different doses of gamma rays from 150 Gy to 350 Gy with 50 Gy intervals to optimize an ideal dose for irradiation. Germination percent of the irradiated seeds ranged from 25.07% (G: 350 Gy) to 80.27% (G: 150 Gy). Probit-based $LD_{50}$ values of P152 were 298.67 Gy, 281.65 Gy and, 268.28 Gy in roll-towel, pro tray and, field experiments, respectively (S2 Fig). Hence, the optimal dose for irradiation of cowpea was fixed at

200, 250, and 300 Gy of gamma rays to induce determinate growth type mutants with TFL in cowpea.

## Identification of TFL mutants

Irradiated ($M_0$) seeds were sown in the field to raise $M_1$ generation. All the survived $M_1$ plants were harvested individually at physiological maturity, and 658 to 1245 plants per dose were advanced to $M_2$ in progeny rows along with the wild type (P152) during Kharif 2019 (June 2019 to August 2019). In the $M_2$ generation, mutant lines were segregated for morphological traits viz., plant height (52.4 to 121.1 cm), days to flowering (41 to 61 days), number of primary branches (3 to 7 branches), number of clusters per plant (4 to 16 clusters), number of pods per plant (11 to 32 pods), peduncle length (14.4 to 28.5 cm), pod length (7.6 to 14.7 cm), number of seeds per pod (7 to 13 seeds), hundred seed weight (7.0 to 15.2 g), days to maturity (72 to 102 days), and single plant yield (8.7 to 20.3 g) across the mutagenic doses. The two TFL mutants (TFLCM-1 and TFLCM-2) identified in $M_2$ generation at 200 Gy were compared with the wild type. These mutants were characterized by TFL/determinate growth habit, more number of pods per cluster, more number of pods per plant, and, increased peduncle length over the wild type. TFL mutants exhibited a reduction in plant height, number of clusters per plant, number of leaves, intermodal length, pod length, number of seeds per pod, hundred seed weight, days to maturity, and single plant yield compared to wild type. Post selfing, all the seeds from the two-TFL mutants (72 and 94 seeds) were harvested individually and forwarded to $M_3$ generation along with the wild type (P152). The *per se* performance and variability parameters for various quantitative characters in $M_3$ generations are given in S3 Fig and S1 Table, respectively. TFL mutants were then forwarded to $M_5$ via $M_4$ generations as progeny rows with no significant difference in performance. In the $M_5$ generation, two selected TFL mutants were designated as TFLCM-1-5 and TFLCM-2-5 and the variability among 11 morphological traits were given in S2 Table. The agronomic performance of the TFL mutants and wild type are presented in Table 2. TFL mutants had purple colour sepals compared to the

**Table 2. Agronomic performance of cowpea terminal flowering mutants and wild type.**

| S.No. | Characters | Indeterminate type (control) | Determinate type (mutants) | | | Mean | CD |
|---|---|---|---|---|---|---|---|
| | | P 152 Wild type | TFLCM-1-5 | TFLCM- 2–5 | TFLCM-MF2 | | |
| 1. | Plant height (cm) | 74.15 | 43.56* | 46.81* | 50.43* | 53.74 | 12.39 |
| 2. | Days to flowering | 49.1 | 46.21* | 45.78* | 47.11* | 47.05 | 1.32 |
| 3. | Mean Inter nodal length (cm) | 5.97 | 3.56* | 4.21* | 3.94* | 4.42 | 0.95 |
| 4. | Number of leaves | 56.49 | 42.15* | 45.32* | 47.54* | 47.88 | 5.49 |
| 5. | No. of primary branches | 4.52 | 4.02* | 3.94* | 4.1* | 4.15 | 0.23 |
| 6. | No. of clusters per plant | 10.93 | 7.75* | 8.12* | 8.03* | 8.71 | 1.33 |
| 7. | No. of pods per plant | 18.32 | 26.21* | 28.84* | 26.47* | 24.96 | 4.09 |
| 8. | Peduncle length (cm) | 19.08 | 31.45* | 30.75* | 31.17* | 28.11 | 5.38 |
| 9. | Pod length (cm) | 11.72 | 8.92* | 8.75* | 8.83* | 9.56 | 1.29 |
| 10. | No. of pods per cluster | 1.81 | 4.56* | 3.97* | 4.77* | 3.78 | 1.21 |
| 11. | No. of seeds per pod | 10.26 | 7.8* | 7.5* | 8.0* | 8.39 | 1.13 |
| 12. | Hundred seed weight (g) | 9.59 | 8.76* | 8.87* | 9.01* | 9.06 | 0.33 |
| 13. | Single plant yield (g) | 16.04 | 14.25* | 14.89* | 15.23* | 15.10 | 0.66 |
| 14. | Days to maturity | 89.5 | 81* | 80.6* | 81.2* | 83.08 | 3.83 |

*Significance @ p(0.05) over wild type.

green wild type. Purple colour pigmentation was found on the tip of the TFL pods while it was absent in wild type. The angles between the peduncle and pod of the TFL mutants were 180º ± 20º, whereas in wild type it was 90º ± 20º (Fig 1).

## MutMap population

The TFL mutants were crossed with the wild type to produce $MF_1$ generation. All the $MF_1$ plants were indeterminate, thus confirming the successful hybridization, as the TFL is governed by a recessive gene. After selfing, the segregating generation $MF_2$ developed is called as MutMap population. The $MF_2$ plants with TFL were designated as TFLCM-MF2 and the mean performance is given in Table 2. In $MF_2$ generation, 1466 indeterminate plant type and 451 plant type were observed. The study on Mendalian inheritance revealed the typical segregating pattern of recessive gene 3:1 ratio with a p-value of 2.220 (Table 3).

## Sequence analysis of *VuTFL1* gene in mutants

*VuTFL1* gene (1291 bp) was subjected to sequencing in six lines viz., two determinate mutants (TFLCM-1-5, TFLCM-2-5), the bulk of determinate segregants of MutMap population (TFLCM-MF2), one indeterminate mutant carried mutation for other traits IDCM-1, and one indeterminate parental type from MutMap population (IDCM-2), and wild type (P152). Multiple sequence alignment (S4 Fig) analysis revealed that five SNP's viz., T insertion at 314 bp, G–A transition at 597 bp, T deletion at 609 bp, A insertion at 968 bp, and A–T transversion at 1196 bp across the mutant lines. Among the five SNPs, four SNPs were in the intronic regions while A to T transversion at 1196 bp was in the exonic region. The predicted protein sequence length was found to be 173 amino acids. The gene *VuTFL1* consists of four exons and 3 introns (Fig 2). A change in amino acid sequence at the 143rd position (N to Y = asparagine to tyrosine) was witnessed in the conserved region in TFL mutants. The amino acid asparagine (N) in the indeterminate plants was substituted with tyrosine (Y) in the determinate/TFL lines (Fig 2).

## *Insilico* protein structure analysis of *VuTFL1* gene

PDB BLAST-based template structure of *1 WKO A* chain was selected, as it showed 100% query coverage and more than 78% sequence identity to the *VuTFL1* gene. Three models of *VuTFL1* were generated using modeller module, satisfying spatial restraints in terms of function violation, least DOPE (Discrete Optimized Protein Energy) score, acceptable geometry, and lowest probability density function the models with least DOPE scores of -18440.62 and -18352.22 were selected as the best for the target comparison of wild and mutant lines, respectively. The structures of wild and mutant lines were superimposed for performing structure-level comparisons. The structure of wild and mutant lines are depicted in red and green colours and the variation in the amino acid are highlighted in yellow and blue colours, respectively (Fig 3). The structures obtained were validated using the Ramachandran plot. The SNP/ 143rd amino acid (N- Y amino acid change) was placed with φ/ψ angles of 141.58 and 111.89, respectively. The secondary structure of that particular amino acid was found to be in the anti-parallel region of the beta-sheet. In the Ramachandran plot, 87.7% of the residues were in the most favoured region and 11.6% of the residues were in additionally allowed regions.

## *Insilico* protein function analysis of *VuTFL1* gene

The effect of amino acid on protein stability/function substitution was predicted using two online tools and the results are depicted in Table 4. A loss in protein function or destabilization

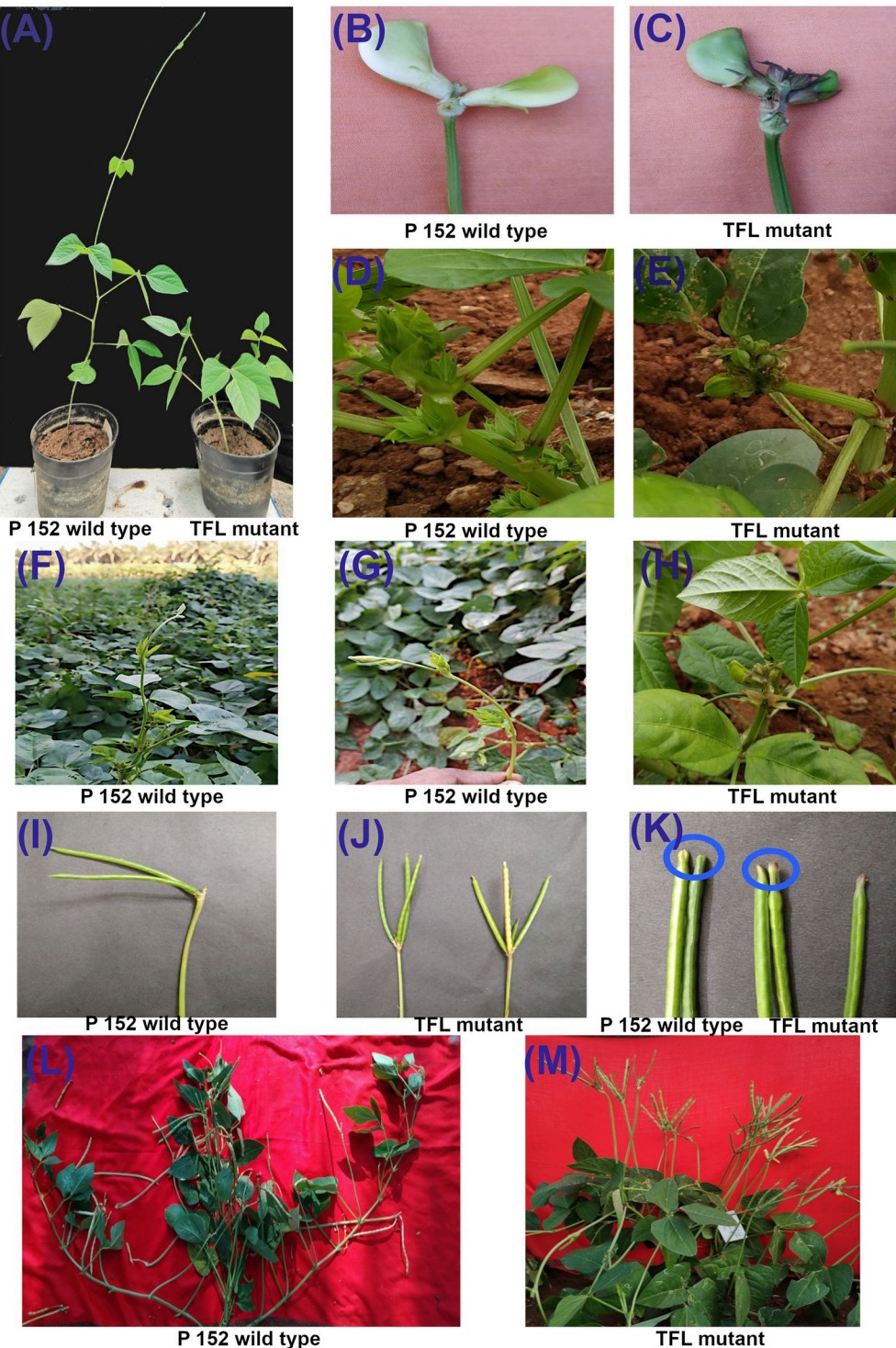

**Fig 1. Phenotypic comparison between the cowpea variety P152 and terminal flowering mutants.** (A) Seedling growth at 25 days after sowing in P152 (wild type) and terminal flowering (TFL) mutant. P152 is tall with tendril development whereas TFL mutant is dwarf with no tendril; (B) Flower of P 152 with greenish white calyx; (C) Flower of TFL mutant with purple pigmentation; (D) P152 wild type with initial of vegetative growth on terminal axis; (E) TFL mutant with initiation of reproductive growth/flowers on terminal axis; (F &G) Tendril formation on the terminal axis of P152; (H) Flower formation on the terminal axis of TFL mutant; (I) Two pods per cluster with an angle of 90º between peduncle and pods; (J) three to five pods per cluster with an angle of 180º between peduncle and pods; (K) P152 with green pod tip while TFL mutant had purple pod tip; (L) Whole plant view of P 152 with indeterminate plant

growth and short peduncle length; (M) Whole plant TFL mutant with determinate growth habit with extended peduncle length.

of protein was predicted as a result of amino acid substitution with a confidence score ranging from 40% (PolyPhen-2) to 82% (PhD-SNP). iMUTANT predicted the deleterious effect with the reliability index 2. The molecular functional disruption was prophesied by MutPred2 and was attributed to the loss of the catalytic site at R139 (P = 0.05) and the gain of sulfation at N143 (P = 0.04). KEGG orthology predicted it to be a phosphatidylethanolamine binding protein (PEBP) involved in the peptidase inhibitors pathway and flower development. KEGG orthology prognosticated that mutant/SNP will affect the biological function of PEBP by negative regulation of flower development. It could also affect the molecular function i.e., transcription coregulatory activity in PEBP. In a nutshell, the SNP obtained due to the mutagenic treatment of gamma rays in the P152 cultivar had a deleterious effect on PEBP and negative regulation of the *VuTFL1* gene.

## Expression profiling of *VuTFL1* gene

The qRT PCR analysis of the *VuTFL1* gene at various growth stages of TFL mutants and wild type is depicted in Fig 4. A slight reduction in the expression level of the *VuTFL1* gene was observed in TFL mutants compared to wild type during the V4 stage of crop growth. In R5/ flowering growth stage, the highest reduction in the expression level of the *VuTFL1* gene was observed in TFL mutants compared to the wild type. It confers that the down-regulation of the *VuTFL1* gene at the flowering stage leads to TFL in cowpea. Furthermore, the difference in the expression level of mutants and wild type is reduced in the R7 growth stage of cowpea. The lowest expression of the *VuTFL1* gene was observed at the R7 stage in both TFL mutants and wild type.

## Discussion

Cowpeas have an indeterminate growth habit, which results in delayed maturity, non-synchronous flowering, photosensitivity, difficulties in intercultural operations, and mechanical harvesting. Most of the Indian landraces and cultivars are indeterminate in nature as this trait was somehow inherited during domestication owing to its multiple pod pickings at regular intervals. Mutation breeding is a highly effective method and, therefore, commonly used in crop breeding to develop determinate plant types. The fruitful accomplishment of any mutation breeding program relies on an intelligent choice of mutagen and dosage optimization. In the present study, a rigorous screening was carried out to optimize the mutagenic dose, and thereby more desirable mutants were obtained. The screening for optimal doses was carried out by three different methods viz., roll-towel method, protray method, and field condition. The calculated $LD_{50}$ values for gamma rays treated P 152 population was around 250 Gy to

**Table 3. Segregation pattern of determinate growth habit in $MF_2$ generation.**

| S. No. | Class | Expected ratio | Observed value (O) | Expected value (E) | Deviation (d = O-E) | $d^2$ / E |
|--------|-------|----------------|--------------------|--------------------|--------------------|-----------|
| 1 | Indeterminate | 3 | 1466 | 1437.75 | 28.25 | 0.555 |
| 2 | Determinate | 1 | 451 | 479.25 | -28.25 | 1.665 |
| | Total | 4 | 1917 | 1917.00 | 0.00 | Cal. $\chi^2$ = 2.220[NS] |

**Note**; *NS–Non Significant p (0.05).

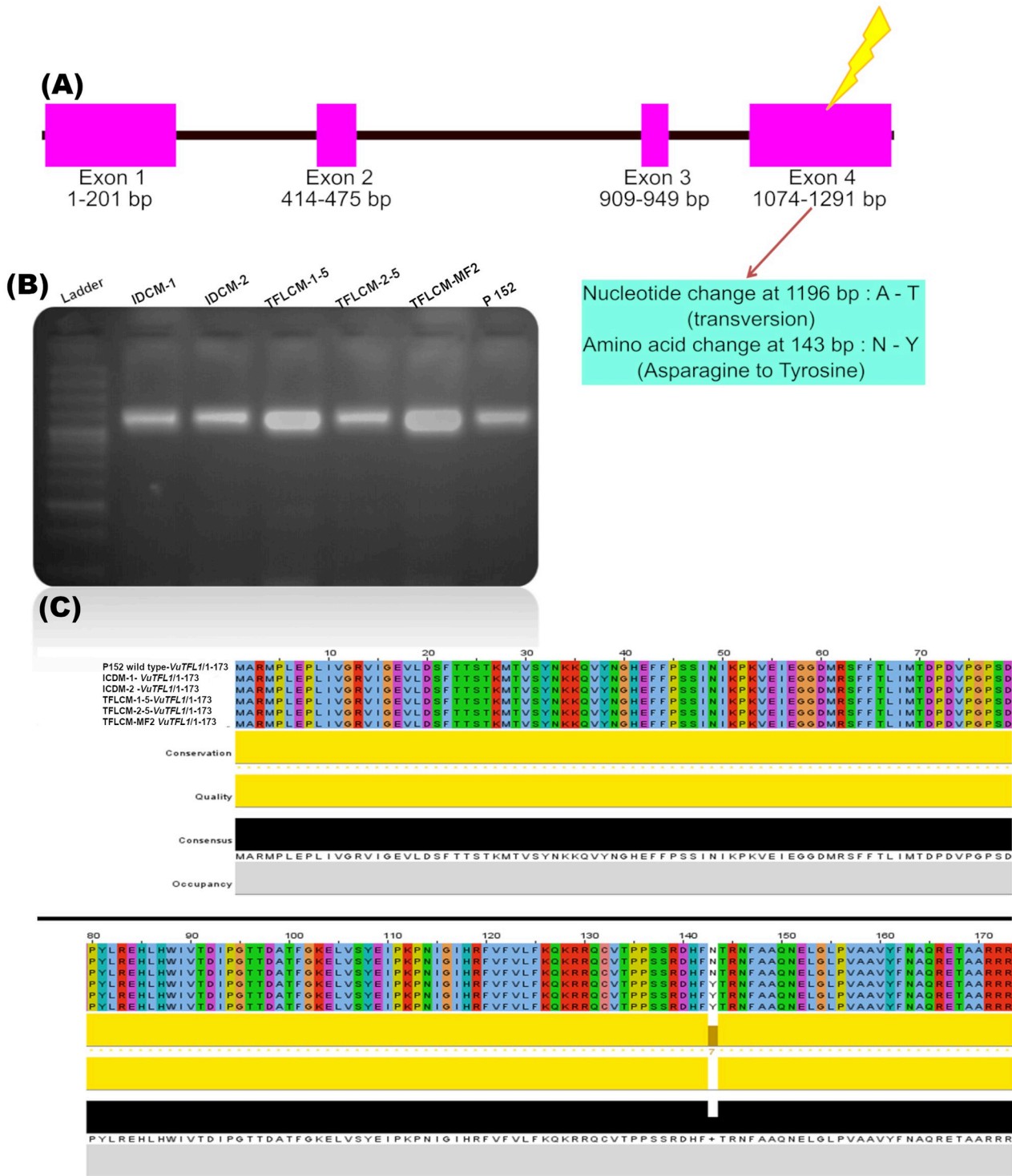

**Fig 2. Cowpea *VuTFL1* gene illustration and sequence alignment.** (A) Illustration of *VuTFL1* gene with single nucleotide variation between P 152 and terminal flowering mutant; B) Gel image of cowpea *VuTF1* gene amplified at 1300 bp with PCR primers; Order of samples: 100 bp ladder; IDCM-1; IDCM-2; TFLCM-1-5; TFLCM-2-5, TFLCM-MF2; P 152; (C) Protein sequences alignment showing single amino acid variation at 143[rd] position (N-Y) of terminal flowering mutants. *Order of alignment*: P 152 wild type; IDCM-1; IDCM-2; TFLCM-1-5; TFLCM-2-5, TFLCM-MF2.

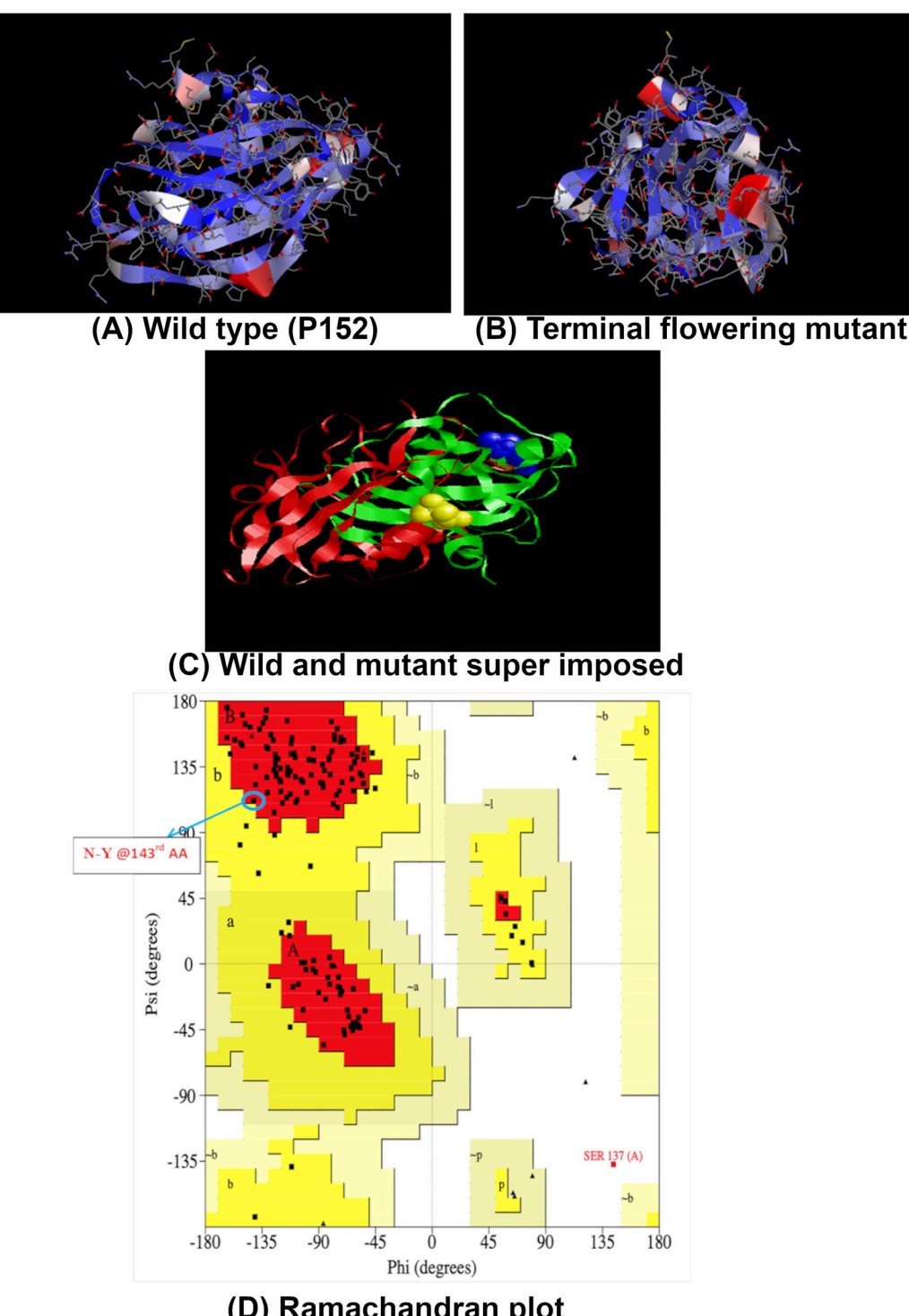

**Fig 3. Protein structure prediction of *VuTFL1* gene in cowpea.** (A) P 152; (B) TFL mutant; (C) Protein structure of wild type and TFL mutant type super imposed; (D) protein structure validation using Ramachandran plot.

**Table 4. *In silico* protein function prediction of cowpea *VuTFL1* gene in mutant lines.**

| S. No. | Software | Prediction | Confidence (%) / Score |
|---|---|---|---|
| 1 | PhD-SNP | Deleterious | 82 |
| 2 | MutPred2 | Deleterious | 70.9% Loss of Catalytic site at R139 (P = 0.05) Gain of Sulfation at N143 (P = 0.04) |

300 Gy. These results are consistent with the reports of Gnankambary, Batieno [48], who described that dose of 250 Gy to 300 Gy in cowpea. In $M_1$ and $M_2$ generation, the effect of mutagens on biological damages based on mortality, sterility, and injury was observed. The considerable variation in the mean and range of various morphological traits in the $M_2$ generation offers great scope for the development of new plant types in cowpea improvement programs. Moreover, most of the viable mutants identified in the $M_2$ generation failed to inherit in later generations as the characters were highly influenced by the environment. In contrast, the TFL mutants identified in the study (200 Gy gamma rays) exhibited a high degree of stability and is governed by a single recessive gene [49,50]. The high heritability coupled with high genetic advance percent means witnessed in the $M_5$ generation of mutants indicated that the traits were fixed.

The TFL mutant also possessed a few modifications in the morphological traits compared to wild plant types. Reduction in morphological traits viz., plant height, duration, hundred seed weight, pod length, number of seeds per pod, internodal length, and number of leaves were observed in TFL mutants as compared to the wild type. These results are similar to previous results Alvarez, Guli [51]; Krylova, Burlyaeva [52]. On the other hand, there was an increase in peduncle length, number of pods per cluster, and number of pods per plant in the TFL mutants as compared to the wild type. This may be due to the fact that the randomness of mutation in some other genes in addition to the *TFL1* gene would have resulted in phenotypic differences. The TFL mutants observed in the present investigation exhibited lesser single plant yield compared to the wild type. Even though the decline in single plant yield was witnessed in TFL/determinate plant type, it is expected to yield on par with the indeterminate/wild type plants owing to its suitability for high-density plantation [53]. Compared to the TFL mutant identified by Dhanasekar and Reddy [5], the mutants identified in the study had higher plant height and longer peduncle length which makes the mechanical harvest easier.

The evolution of mutation breeding over the years from conventional induced mutagenesis to TILLING, genome editing, and MutMap is mind-boggling. MutMap is a recent and advanced technology that combines mutation breeding, hybridization, and second and third-generation sequencing technologies. Mutmap had the advantage of detecting the QTLs, single nucleotide polymorphism (SNPs), and candidate genes associated with trait modifications [54]. The TFL mutants (female) identified in the present study were crossed with the wild type (P 152 as male). In the $MF_1$ generation, plants showing indeterminate types (True $F_1$'s) were selected and allowed to self-pollinate. The segregation of 3 indeterminate: 1 determinate types in $MF_2$ generation suggested the trait to be under the governance of a single recessive gene for TFL in cowpea. [55] reported that *VuTFL1* is the candidate gene responsible for the TFL type in cowpea, and also this gene family was well established in *Arabidopsis* [56], peas [28] and soybean [57]. In the present study, the *VuTFL1* gene was sequenced in mutants and wild-type plants. The multiple sequence alignments of the *VuTFL1* gene (1291 bp) revealed five SNPs across the mutant lines of which four SNPs were found to be in different introns, while A to T transversion at 1196 bp was in the exon of all TFL mutants. All the TFL mutants obtained from the study were derived from a single $M_1$ plant. Hence, the same SNP might be obtained

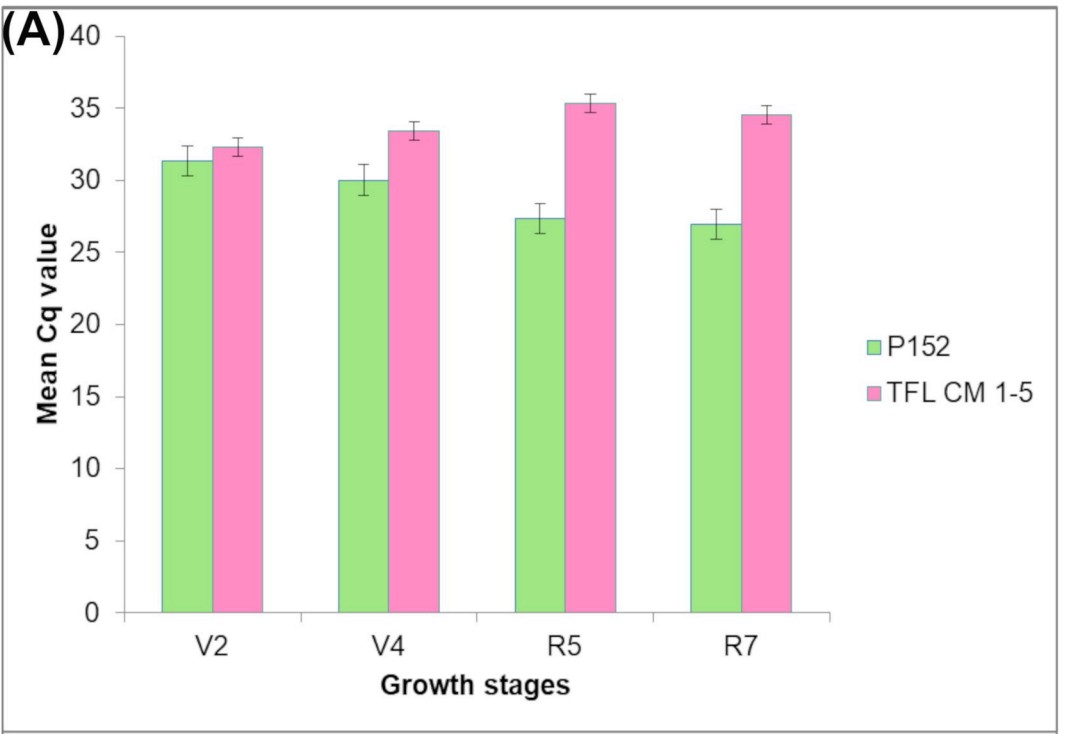

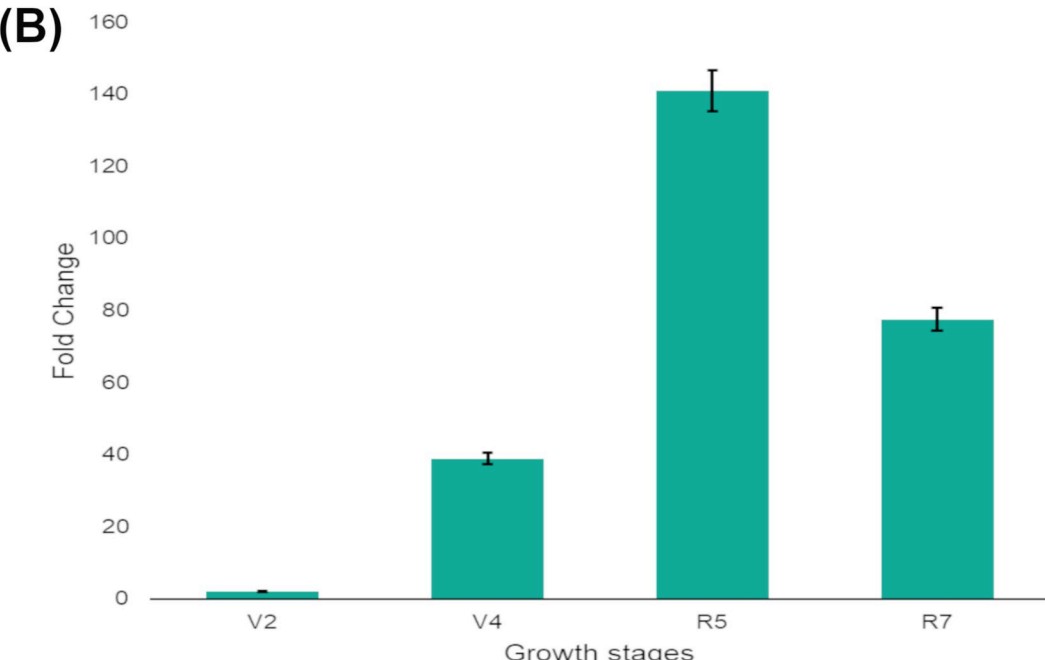

**Fig 4. Expression profiling of *VuTFL1* gene at various growth stages in P152 and TFL mutant.** (A) Comparison of mean Cq values of TFL mutant and wild type (P152) cowpea. (B) Fold change in expression ($2^{-\Delta\Delta Ct}$) of cowpea TFL mutant and wild type (P152) at four growth stages. Error bars represent experimental standard deviations.

in all TFL mutants. Likewise, three-point mutations at the same position of *PsTFL1a* were observed in two determinate mutants in peas by Foucher [22]. Similarly, the mutation on the fourth exon of *TFL1* gene homologs was reported in lablab and soybean by Kaldate, Patel [58] and Liu, Watanabe [57]. On the other hand, Dhanasekar and Reddy [5] reported a point mutation at 1176 bp of *VuTFL1* that led to the development of TFL mutants in cowpea. Both *TFL1* and *FT* genes were found to be highly conserved across the organisms [59,60]. The point mutation on highly conserved regions might alter the protein sequence and consecutively results in a phenotypic change. Hence, it is worth estimating the effect of mutation on proteins [61]. Translational and functional genomics in cowpea is restricted because of its recalcitrant nature during genetic transformation. Hence, the bioinformatics tools offer a great deal of *in silico* clarifications on protein function and structure [62]. In the current study, Soft berry's FGE-NESH predicted the TFL protein in mutants and control to be 173 amino acids long. An amino acid change from asparagine (N) to tyrosine (Y) was found on the 143$^{rd}$ position of protein and that corresponds to the fourth exon of TFL mutants. Similarly, Ahn, Miller [23] reported that a functional mutation in the fourth exon provides the greater plasticity for the conversion of indeterminate to determinate plant types.

Although genome sequencing provides the variation at nucleotide and amino acid sequence levels, the biological function of that variation can be retrieved through the protein structure and function analysis [63]. The 3D protein structure of an unknown target was predicted based on one or more proteins of known structures called templates [64]. PDB BLAST retrieved the template structure of the *1wkoA* chain, as it has 100% query coverage and more than 78% sequence identity to the *VuTFL1* gene. Likewise, Danilevskaya, Meng [65] employed the *1wkoA* chain as a template during the structure prediction of *TFL/FT* genes in maize. The protein model was built using MODELER Accelrys discovery studio with the lowest DOPE score and protein structure of wild type and TFL mutants were super imposed and compared. Mutation in TFL type leads to the modification of the external loop of beta-sheet with less than one percent of residues in the disallowed region. The modification in secondary structure and the external adjacent loop formation were noticed due to amino acid residues 139–144 of the *TFL1* gene [66]. These modifications in the external loop of beta-sheets would have suppressed the expression of a *VuTFL1* gene and resulted in TFL/determinate mutants. KEGG orthology predicted the *VuTFL1* gene belonged to phosphatidyl ethanolamine binding protein (PEBP). It is an ancient protein family found across the biosphere and plays a vital role in flower regulation [67].

Understanding the effect of point mutation on the phenotype of the plant via alteration of amino acid sequence and function is of significant interest in plant breeding. The evolutionary conservation of an amino (or nucleic) acid position is greatly influenced by its structural and functional relevance; rapidly evolving positions are unstable, whereas slowly evolving positions are conserved. Hence, the studies on conservation assessment of position highlight the importance of each position for the structure or function of the proteins. Flanagan, Patch [61] reported that 88% of the mutations disrupt highly conserved residues. In the present study, protein stability/function was predicted using eight online tools. Sequence homology-based method to predict the protein damage triggered by random mutations can be applied across organisms [68]. A loss in protein function, destabilization of protein, loss of catalytic site, and gain of sulfation were witnessed as a result of amino acid substitution in various prediction tools with a confidence score of 40% to 82%. A mutation in *TFL1* with a functional loss would result in the conversion of SAM into a terminal flower [21,22]. KEGG orthology prognosticated that mutant/SNP will affect the biological function of PEBP by negative regulation of the *TFL1* gene. It also affects molecular function i.e., transcription co-regulator activity in PEBP. The feasibility of mutation in inducing the TFL in cowpea has been previously manifested by

Dhanasekar and Reddy [5] and Pandey and Dhanasekar [69]. The *TFL1* gene plays a major role in determining the plant architecture. Overexpression of the *TFL1* boosts the SAM and leads to indeterminate growth habit. On the other hand, loss of function exhausts the SAM and promotes TFL in crop plants [70]. In the present study, expression analysis revealed the down-regulation of *VuTFL1* gene at V4, R5 and R7 stages of TFL cowpea mutants. The mutation of *VuTFL1* gene leads to down-regulation or loss of function of PEBP and aligns well with the recessive inheritance of determinate growth habit [28]. Hence, loss of function mutation of *VuTFL1* gene and PEBP induced through 200 Gy of gamma rays in the study arrested the SAM and induced TFL growth habit in cowpea mutants.

In conclusion, the identification of allelic variations within genes related to growth habits holds the potential to illuminate the molecular mechanisms underlying the crop ideotype. This comprehension, in turn, has the potential to lay the groundwork for modifying growth patterns. The findings of this study suggest that the loss of function in the VuTFL1 gene in cowpea adversely affects vegetative growth while promoting the TFL growth habit. The stable TFL mutants generated in this research hold promise for advancing innovative determinate cowpea cultivars, which could address challenges in intercultural and other cultivation practices while facilitating mechanical harvesting. This study serves as an exemplary demonstration of successfully integrating traditional induced mutagenesis with genomics to reshape plant architecture, thereby facilitating the development of nutritious and climate-resilient crops. Furthermore, our investigation introduces the viability of enhanced MutMap techniques within pulse crops for the first time. This exploration also offers the potential to extend the application of this technique to other enhancement initiatives involving pulse crops.

## Supporting information

**S1 Fig. Schematic representation for development of terminal flowering mutants and Mut-Map population in cowpea.**
(DOCX)

**S2 Fig. Mutagenic dosage optimization of gamma rays in P152 cowpea cultivar through $LD_{50}$.**
(DOCX)

**S3 Fig. Frequency distribution of various morphological traits in $M_3$ generation of cowpea cultivar P152 generated by gamma irradiation.**
(DOCX)

**S4 Fig. Multiple sequence alignment of *VuTFL1* gene in cowpea mutants.**
(DOCX)

**S1 Table. Mean, variability and heritability estimates of quantitative traits in $M_3$ generation of cowpea cultivar P152 generated by gamma irradiation.**
(DOCX)

**S2 Table. Variability parameters for 11 morphological traits in $M_5$ generation of cowpea mutants.**
(DOCX)

## Author Contributions

**Conceptualization:** Vijayakumar Eswaramoorthy, Thangaraj Kandasamy.

**Data curation:** Vijayakumar Eswaramoorthy.

**Formal analysis:** Vijayakumar Eswaramoorthy.

**Funding acquisition:** Thangaraj Kandasamy.

**Investigation:** Vijayakumar Eswaramoorthy.

**Methodology:** Vijayakumar Eswaramoorthy, Senthil Natesan.

**Project administration:** Thangaraj Kandasamy, Kalaimagal Thiyagarajan, Souframanien Jegadeesan.

**Resources:** Vijayakumar Eswaramoorthy, Kalaimagal Thiyagarajan, Vanniarajan Chockalingam, Souframanien Jegadeesan, Senthil Natesan, Raveendran Muthurajan.

**Software:** Vijayakumar Eswaramoorthy, Senthil Natesan.

**Supervision:** Thangaraj Kandasamy, Kalaimagal Thiyagarajan, Vanniarajan Chockalingam, Souframanien Jegadeesan, Senthil Natesan, Jeyakumar Prabhakaran, Ramji Singh, Raveendran Muthurajan.

**Validation:** Vijayakumar Eswaramoorthy, Jeyakumar Prabhakaran, Ramji Singh.

**Visualization:** Vijayakumar Eswaramoorthy.

**Writing – original draft:** Vijayakumar Eswaramoorthy, Karthikeyan Adhimoolam.

**Writing – review & editing:** Vijayakumar Eswaramoorthy, Thangaraj Kandasamy, Souframanien Jegadeesan, Karthikeyan Adhimoolam.

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
