## [Decision Letter · Decision Letter 0]

14 Aug 2023

PONE-D-23-01909Characterization of terminal flowering cowpea (Vigna unguiculata (L.) Walp.) mutants obtained by induced mutagenesis digs out the loss-of-function of phosphatidylethanolamine-binding proteinPLOS ONE

Dear Dr. Kandasamy,

Thank you for submitting your manuscript to PLOS ONE. After careful consideration, we feel that it has merit but does not fully meet PLOS ONE’s publication criteria as it currently stands. Therefore, we invite you to submit a revised version of the manuscript that addresses the points raised during the review process.

In general, the two reviewers feel the study is technically sound. Please address the minor comments. 

We look forward to receiving your revised manuscript.

Kind regards,

Jianhong Zhou

Staff Editor

PLOS ONE

Journal Requirements:

"Research work was supported by the "Board of Research in Nuclear Sciences (BRNS)" by the Bhabha Atomic Research Centre (BARC), Trombay, India. Grant number: No.35/14/18/2018-BRNS/10396 dt. 25.05.2018."

Reviewers' comments:

Reviewer's Responses to Questions

**Comments to the Author**

1. Is the manuscript technically sound, and do the data support the conclusions?

Reviewer #1: Yes

Reviewer #2: Yes

2. Has the statistical analysis been performed appropriately and rigorously? 

Reviewer #1: Yes

Reviewer #2: Yes

3. Have the authors made all data underlying the findings in their manuscript fully available?

Reviewer #1: Yes

Reviewer #2: Yes

4. Is the manuscript presented in an intelligible fashion and written in standard English?

Reviewer #1: Yes

Reviewer #2: Yes

5. Review Comments to the Author

Reviewer #1: The article "Characterization of terminal flowering cowpea (Vigna unguiculata (L.) Walp.) mutants obtained by induced mutagenesis digs out the loss-of-function of phosphatidylethanolamine-binding protein" presents an interesting study on the development and characterization of determinate habit mutants with terminal flowering (TFL) in cowpea. The mutants exhibited desirable traits such as TFL growth habit, longer peduncles, erect pods, increased number of pods per cluster, and early maturity. A significant finding of the study is the identification of a single nucleotide transversion in the VuTFL1 gene of the determinate mutants resulting in a loss-of-function mutation. and the MutMap population. The identification and characterization of the loss-of-function mutation in the PEBP gene contribute to our understanding of the underlying molecular mechanisms controlling these traits.

The article is generally well-written and organized, presenting the methodology, results, and implications of the study in a clear and concise manner. The article will make a valuable contribution to the scientific literature in the field of crop genetics and breeding; therefore, I recommend for publication in PlosONE.

Reviewer #2: Dear Authors,

This is a very interesting study on cowpea. However, I have two comments. Firstly, the take home message is not very clear. Secondly, the quality of the figures are not up to the mark. Many of the figures are blurred and illegible. Therefore, I strongly recommend to address these two comments and resubmit.

6. PLOS authors have the option to publish the peer review history of their article (what does this mean?). If published, this will include your full peer review and any attached files.

Reviewer #1: No

Reviewer #2: **Yes: **Dr. Soumya Ghosh

---

## [Author Response · Author response to Decision Letter 0]

26 Aug 2023

Response to editor and reviewer comments

Editor comments 

Thank you for submitting your manuscript to PLOS ONE. After careful consideration, we feel that it has merit but does not fully meet PLOS ONE’s publication criteria as it currently stands. Therefore, we invite you to submit a revised version of the manuscript that addresses the points raised during the review process. In general, the two reviewers feel the study is technically sound. Please address the minor comments. Please submit your revised manuscript by Sep 28 2023 

Response 

Thank you very much for handling our manuscript and giving us the opportunity to submit a revised draft of our manuscript. We appreciate the time and effort you and the reviewers dedicated to providing valuable feedback on our manuscript. We are grateful to the reviewers for their insightful comments. According to the comments, we have addressed all the issues and revised the manuscript carefully. The changes made in the manuscript are marked with blue color/tracks. We hope the revised version is now suitable for publication We look forward to hearing from you in due course regarding our submission and responding to any further questions and comments you may have. 

Reviewer #1: 

Comments

The article "Characterization of terminal flowering cowpea (Vigna unguiculata (L.) Walp.) mutants obtained by induced mutagenesis digs out the loss-of-function of phosphatidylethanolamine-binding protein" presents an interesting study on the development and characterization of determinate habit mutants with terminal flowering (TFL) in cowpea. The mutants exhibited desirable traits such as TFL growth habit, longer peduncles, erect pods, increased number of pods per cluster, and early maturity. A significant finding of the study is the identification of a single nucleotide transversion in the VuTFL1 gene of the determinate mutants resulting in a loss-of-function mutation. and the MutMap population. The identification and characterization of the loss-of-function mutation in the PEBP gene contribute to our understanding of the underlying molecular mechanisms controlling these traits. The article is generally well-written and organized, presenting the methodology, results, and implications of the study in a clear and concise manner. The article will make a valuable contribution to the scientific literature in the field of crop genetics and breeding; therefore, I recommend for publication in PlosONE.

Response: On behalf of all the authors, I am writing to express our sincere gratitude for your thorough review and invaluable feedback to our manuscript. Your positive evaluation of the work is truly motivating and reassuring. It gives us great pleasure to know that our collaborative efforts have culminated in this positive outcome. We also appreciate the time you have taken for the revision process and your valuable comments.

Reviewer #2: 

Comments

Dear Authors, This is a very interesting study on cowpea. However, I have two comments.

Response: Thank you for your thoughtful review of our manuscript. We sincerely appreciate the time and effort you have invested in evaluating our work. Your feedback is invaluable to us as we strive to enhance the quality and impact of our research.

Comment 1: Firstly, the take home message is not very clear:

Response 1: Thanks for your kind suggestions. We have rephrased our conclusion to ensure that the main findings and implications were presented more effectively. Kindly check the revised manuscript Abstract and discussion sections. 

Comment 2: Secondly, the quality of the figures are not up to the mark. Many of the figures are blurred and illegible.

Response 2: We apologize for the quality issues you identified in the figures. We have carefully reviewed and improved the resolution of all images using the PACE software, as per the journal's recommendation, to ensure standard of the figures.

---

## [Decision Letter · Decision Letter 1]

27 Nov 2023

Characterization of terminal flowering cowpea (Vigna unguiculata (L.) Walp.) mutants obtained by induced mutagenesis digs out the loss-of-function of phosphatidylethanolamine-binding protein

PONE-D-23-01909R1

Dear Dr. Kandasamy,

We’re pleased to inform you that your manuscript has been judged scientifically suitable for publication and will be formally accepted for publication once it meets all outstanding technical requirements.

Kind regards,

Sumit Jangra, Ph.D.

Academic Editor

PLOS ONE

Additional Editor Comments (optional):

Reviewers' comments:

Reviewer's Responses to Questions

**Comments to the Author**

1. If the authors have adequately addressed your comments raised in a previous round of review and you feel that this manuscript is now acceptable for publication, you may indicate that here to bypass the “Comments to the Author” section, enter your conflict of interest statement in the “Confidential to Editor” section, and submit your "Accept" recommendation.

Reviewer #1: All comments have been addressed

Reviewer #3: All comments have been addressed

2. Is the manuscript technically sound, and do the data support the conclusions?

Reviewer #1: Yes

Reviewer #3: Yes

3. Has the statistical analysis been performed appropriately and rigorously? 

Reviewer #1: Yes

Reviewer #3: Yes

4. Have the authors made all data underlying the findings in their manuscript fully available?

Reviewer #1: Yes

Reviewer #3: Yes

5. Is the manuscript presented in an intelligible fashion and written in standard English?

Reviewer #1: Yes

Reviewer #3: Yes

6. Review Comments to the Author

Reviewer #1: The authors have satisfactorily revised the manuscript. The manuscript provides valuable information on the development of determinate cowpea mutants and their genetic characterization.

Reviewer #3: Dear Author

All the requests of the reviewers have been done and therefore it is acceptable.

Regards

7. PLOS authors have the option to publish the peer review history of their article (what does this mean?). If published, this will include your full peer review and any attached files.

Reviewer #1: No

Reviewer #3: No

---

## [Editor Report · Acceptance letter]

4 Dec 2023

PONE-D-23-01909R1 

Characterization of terminal flowering cowpea (*Vigna unguiculata* (L.) Walp.) mutants obtained by induced mutagenesis digs out the loss-of-function of phosphatidylethanolamine-binding protein 

Dear Dr. Kandasamy:

I'm pleased to inform you that your manuscript has been deemed suitable for publication in PLOS ONE. Congratulations! Your manuscript is now with our production department. 

Kind regards, 

on behalf of

Dr. Sumit Jangra 

Academic Editor

PLOS ONE